# Use of High-Resolution Ultrasound in Characterizing the Surface Topography of a Breast Implant

**DOI:** 10.3390/medicina59061092

**Published:** 2023-06-05

**Authors:** Yang-Hee Kim, Dong-Wook Park, Keun-Yeong Song, Hyung-Guhn Lim, Jeong-Pil Jeong, Jae-Hong Kim

**Affiliations:** 1Department of Surgery, Kangwon National University, Chuncheon 24289, Republic of Korea; kyh1524@kangwon.ac.kr; 2Venus Women’s Clinic, Seoul 07727, Republic of Korea; marvelous@naver.com; 3Department of Breast Surgery, Gwangju Suwan Hospital, Gwangju 62247, Republic of Korea; sky7939@hanmail.net; 4Department of Radiology, Gwangju Suwan Hospital, Gwangju 62247, Republic of Korea; pmclimhg@hanmail.net; 5Samsungyubang Breast Clinic, Busan 48104, Republic of Korea; aha2249@naver.com; 6The W Clinic, Seoul 06038, Republic of Korea

**Keywords:** breast implants, ultrasonography, topography, medical, medical record, lymphoma

## Abstract

*Background and Objectives*: With the emergence of breast implant-associated anaplastic large cell lymphoma (BIA-ALCL), it has become necessary to identify the implant shell type patients have received. Therefore, an immediate, reliable method for identifying a breast implant shell type is essential. Evidence-based research and applying a real-world technique that identifies the surface topographic information of the inserted breast implants, without surgery, has become of paramount importance for breast implant physicians. *Methods and Materials*: A review of the medical records of 1901 patients who received 3802 breast implants and subsequently received an ultrasound-assisted examination was performed. All patients received not only a breast cancer examination but also a high-resolution ultrasonography (HRUS) assisted examination of the device at a single center between 31 August 2017 and 31 December 2022. *Results*: Most patients had breast implants within 10 years (77.7%) of the examination. Of the 3802 implants screened, 2034 (53.5%) were identified with macro-textured shell topography in ultrasonography. A macrotextured shell type implant was used in 53.5% of cases and a smooth type in 42.7% of cases. Seventy-three (1.9%) breast implant shell types could not be identified due to ruptures. However, 250 breast implant shell types could be identified despite rupture cases (6.5%). *Conclusions*: HRUS was found to be a useful and reliable image modality for identifying various surface shell types of breast implants. The shell type information would be helpful to patients who lack information about their breast implants and are concerned about BIA-ALCL.

## 1. Introduction

Breast implants were first manufactured and then sold by the Dow Corning Corporation in 1963. Since then, there have been great changes made in the topography of the device [1]. In 1968, a textured breast implant was first developed [2].

Implant surface texture has a relationship with capsular contracture (CC) and breast implant-associated anaplastic large cell lymphoma (BIA-ALCL) [2,3,4,5,6,7,8,9,10]. After implantation, the device is recognized as a foreign body and initiates an immune response with the formation of a collagen fiber capsule [7,8].

CC is a major complication characterized by pain, discomfort, and asymmetry after the occurrence of capsule tightening and hardening around a breast implant [11].

In addition, BIA-ALCL is a rare T-cell non-Hodgkin lymphoma that develops around a breast implant with a textured surface in both esthetic and reconstructive surgery [12], characterized as CD30-positive and anaplastic lymphoma kinase (ALK)-negative. 

In 1997, Dr. Keech reported the first case of BIA-ALCL [13]. Since then, 952 patients with BIA-ALCL have been reported worldwide, including 32 deaths attributed to BIA-ALCL. 

Most cases have demonstrated an indolent course after explantation, capsulectomy, chemotherapy, and radiotherapy, but some cases are refractory to standard therapy, ultimately causing death [14,15,16]. 

Peri-implant late seromas are the most common, where pericapsular masses, and a skin rash is the predominant symptoms [17].

The mean age of diagnosis is 47 years, with the onset being 7.5 years following implant surgery in both esthetic and reconstructive surgeries [18]. 

Three cases were reported in Korea, which led to an Allergan Biocell (Allergan Inc., Irvine, CA, USA) to a issue a recall in 2019 [10,19]. 

After the recall, only micro/nanotextured and smooth breast implants were commercially available in the market today [19,20,21]. 

There are four types of shell surface topographies of breast implant shells: macrotexture, microtexture, nanotexture, and smooth.

Due to the poor management of breast implant information for patients receiving implant-based mammaplasty for esthetic purposes, it was difficult to ascertain the number of patients that received a textured devices manufactured by Allergan Biocell (Allergan Inc., Irvine, CA, USA) when the first Korean case of BIA-ALCL occurred in 2019. This was an issue during an inspection and audit of the Korean National Assembly by the Health and Welfare Committee in 2019, which remains unresolved to this day [22,23].

With the emergence of BIA-ALCL, patients have become aware of the importance of knowing the correct information about implanted devices. Due to the characteristics of local clinics, however, some patients have encountered the problem as it is impossible to clarify the information about breast implants because of doctor’s frequent movement, closure of business, and disposal of medical records after 10 years. 

As described above, no methods were available for identifying information about breast implants in patients. It has become important to identify the manufacturer without surgery.

More importantly, accurate methods for identifying a macrotextured breast implants with a causal relationship with BIA-ALCL have increased interest in clinical settings. 

Many basic studies have recently proposed methods for classifying their surface topography in vitro. However, those studies are insufficient in immediately resolving real-world problems. This study was therefore conducted to classify diverse types of surface topographies using HRUS, thus attempting to resolve these issues in clinical settings.

## 2. Materials and Methods

This retrospective study was approved by the Internal Institutional Review Board of the Korea National Institute of Bioethics Policy (IRB No. P01-202202-01-009), which waived the requirement for the informed consent of medical records, including images and patient characteristics.

All procedures described herein were performed in accordance with the 1964 Declaration of Helsinki and its later amendments or comparable ethical standards.

## 3. Study Population

Ultrasonographic images of 3802 breast implants in 1901 women were retrospectively investigated for this study. All patients received not only a breast cancer examination but also a HRUS-assisted examination of the device at a center between 31 August 2017 and 31 December 2022. 

## 4. Breast Implant Ultrasonography Protocol

All US examinations were performed using an Aplio i600 (Canon Medical Systems, Otawara, Tochigi, Japan) system with a 7–18 MHz linear transducer.

Ultrasonographic images were retrospectively reviewed by a breast surgeon with 13 years of experience in performing breast implant ultrasonographies.

This study examined cross-sectional images of shell surface topography in the lower breast area (Figure 1). 

Cross-sectional images of sample breast implants were examined using a stereomicroscope (Olympus SZ61; Olympus Optical Co., Tokyo, Japan), and its findings were photographed using the TUCSEN H series digital camera (Fuzhou Tucsen Photonics Co., Fuzhou, Fujian, China) at magnification rates of 400× and 1000×. 

## 5. Breast Implant Shell Topography

There are four types of shell surface topography: macrotexture, microtexture, nanotexture, and smooth. 

Representative gross images of the four types of breast implant shell surfaces are shown in Figure 2.

## 6. Image Analysis

In the current study, all breast implants were obtained from eight manufacturers (Groupe Sebbin SAS, Boissy-l’Aillerie, France; HansBiomed Co., Ltd., Seoul, Republic of Korea; Establishment Labs Holdings Inc., Alajuela, Costa Rica; GC Aesthetics PLC, Apt Cedex, France; Allergan Inc., Irvine, CA, USA; Mentor Worldwide LLC, Santa Barbara, CA, USA; Polytech Health & Aesthetics, Dieburg, Germany; and Silimed Inc., Rio de Janeiro, Brazil) and were sold as 14 different brands (macrotextured, microtextured, nanotextured and smooth devices) in total. 

N/A means there was no data available because the device was rarely used in Korea as it was not manufactured in or imported to Korea. 

A comparative analysis of the microscopic cross-sectional views and shell surface topographic images were performed in vivo and in vitro using HRUS. In Korea, breast implants are commercialized as macrotextured, microtextured, and smooth (including nanotextured) devices by eight manufacturers. Thus, a total of 14 brands of breast implants are available in the market (Table 1). Breast implants examined herein were from eight manufacturers including the following: Allergan, Mentor, Silimed, Sebbin, Polytech, Motiva, Eurosilicone, and HansBiomed. The shell surface topographies of sonographic image in 10 macro-textured and micro-textured breast implants from seven manufacturers was analyzed, as were four nanotextured/smooth breast implants from four manufacturers (Figure 3, Figure 4, Figure 5 and Figure 6).

Surgeons from the USA commonly encounter the Food and Drug Administration-approved breast implants from Allergan, Mentor and Silimed. Surgeons from the EU meanwhile commonly encounter the CE-approved breast implants manufactured in the UK, France, and Germany. In Korea, the both CE- and FDA-approved breast implants have been commercialized with approval by the KFDA. Thus, diverse types of breast implants have been used in Korea. This explains why diverse information can be obtained about breast implants.

## 7. Results

Table 1 shows available shell surface types of breast implants from 8 manufacturers.

Table 2 shows the characteristics of the 1,901 patients. Most examinations (77.8%) were under 10 years of implant surgery. Additionally, the mean age of the patients was 36 years. 

Table 3 shows the characteristics of 3802 breast implants. Most cases involved an esthetic silicone breast implant-based mammaplasty (99.8%) via a trans-axillary incision (65.8%). Silicone breast implants were used in most cases (95.1%).

Macro-textured shell type implants accounted for 53.5% of breast implants and smooth type accounted for 42.7%. Seventy-three (1.9%) breast implant shell types couldnot be identified due to rupture. However, 250 breast implant shell types were identified even in ruptured cases (6.5%). 

Various manufacturers were identified using ultrasonography, such as Groupe Sebbin SAS, Boissy-l’Aillerie, France; HansBiomed Co., Ltd., Seoul, Republic of Korea; Allergan Inc., Irvine, CA, USA; Mentor Worldwide LLC, Santa Barbara, CA, USA; and others (Table 3).

Ultrasonography was a useful imaging modality to identify the various shell surface types of the breast implants, and it was compared with a gross cross-sectional views using a light microscope.

These are macro-textured topographic images of breast implants from seven manufacturers; they were observed using a light microscope (Olympus SZ61; Olympus Optical Co., Tokyo, Japan) at a magnification rate of 400×.

Of eight manufacturers, Motiva Korea has imported only microtextured breast implants. Therefore, macro-textured breast implants were analyzed from only seven manufacturers. 

There are three methods to manufacturing a macrotextured breast implant. A macrotextured device can be classified as an embossed or engraved types [5,24,25]. 

In the current study, Allergan Biocell, Sebbin, GC Aesthetics Eurosilicone macrotextured, and HansBiomed macrotextured breast implants belong to a group of salt loss manufactured implants. 

Mentor Siltex corresponds to a polyurethane molded implant surface. Furthermore, Motiva SilkSurface corresponds to a chuck-molded manufactured implants. Finally, Silimed True and Polytech POLYtxt are classified as alternatively textured implants [5].

Representative embossed breast implants include Mentor, Silimed, and Polytech, the ultrasonographic shell surface topographic images of Figure 3D–F show Mentor, Silimed, and Polytech, repectively. 

Representative engraved breast implants include Allergan, Sebbin, HansBiomed, and Eurosilicone, the microscopic and ultrasonographic images in vivo and in vitro of which are well displayed in Figure 4E–H. 

The surface shell types of all seven macro-textured breast implants differed in high-resolution ultrasonography, so, the macro-textured shell type can be used to identify the device manufacturer.

Microtextured breast implants approved for clinical use in Korea include Motiva SilkSurface, Sebbin microtexture, HansBiomed BellaGel SmoothFine and GC Aesthetics Eurosilicone Crystalline Round Collection. Of these, the Motiva shell surface is categorized technically as nanotexture. 

Motiva, Sebbin and HansBiomed were observed at a magnification rate of 1000× and Eurosilicone was observed at a magnification rate of 400× with light microscopy and compared with in vivo and in vitro ultrasonographic shell surface topography (Figure 5). 

Using high-resolution ultrasonography, Motiva, Sebbin, and HansBiomed nano/microtextured sonographic images are well differentiated from the macro-textured breast implants. 

The microscopic and ultrasonographic cross-sectional view of GC Aesthetic Eurosilicone microtextured breast implants are close to that of a macro-textured breast implant. However, the in vivo US findings are closer to micro-textured breast implants. So, it is well differentiated from macrotextured breast implants, as well. 

A/D, B/E, and C/F of Figure 3 are light microscopic and HRUS cross-sectional views of Mentor Siltex, Silimed True, and Polytech POLYtxt macrotextured breast implants, respectively. Two hyperechoic lines in the box are shells. The upper hyperechoic line is the upper layer that is indicative of a textured topography. Each breast implant has its own unique US image, which allows identification of the corresponding manufacturer.

A/E, B/F, C/G, and D/H of Figure 4 are light microscopic and HRUS cross-sectional views of Allergan Biocell, Sebbin macro-texture, HansBiomed macro-texture and GC Aesthetics macro-texture, respectively. These three manufacturers share the common characteristics: a macro-textured breast implant manufactured using a representative engraved method. Because of this, they show similar cross-sectional images. Two hyperechoic lines in the box are shells. The upper hyperechoic line is the upper layer that is indicative of a textured topography. The difference in the number of shell layers and thickness of the lower hyperechoic line may be a clue to the identification of the manufacturer [2].

A, E, and I of Figure 5 are light microscopic, in vitro and in vivo US images of the Motiva SilkSurface implant, respectively, corresponding to a nanotextured breast implant. Two upper hyperechoic lines of the shell in the box may be a clue to identify the manufacturer or macro-textured devices. However, it is difficult to differentiate between nanotextured and smooth breast implants. This might be due to a nano-textured structure of <5 μm.

B, F, and J of Figure 5 are images of the BellaGel SmoothFine microtexture from HansBiomed. Based on the upper hyperechoic line indicating the shell, it can be clearly differentiated from a macrotextured breast implant but cannot be well differentiated from a smooth device.

C, G, and K of Figure 5 are images of Sebbin microtexture. Its upper hyperechoic line can be clearly differentiated from a macrotextured breast implant. However, it cannot be well differentiated from a smooth breast implant. Nevertheless, the identification of the typical structure of the shell may lead to the manufacturer’s confirmation in the case of the Sebbin micro-texture shell type.

D, H, and L of Figure 5 are images of the Eurosilicone Round Collection Crystalline microtexture ES line product from GC Aesthetics, which show slightly different features from microtextured and nanotextured breast implants. Based on the upper hyperechoic line, it can be differentiated from a macrotextured breast implant and other types of microtextured or smooth devices. Its unique images may also serve as a clue to the identify the manufacturer’s identity.

Smooth surface shell types are easily differentiated from macro-textured onee. The smooth types of implants from Allergan, Mentor, and Sebbin are the most common breast implants found in Korea. Microscopic, in vivo and in vitro US images are displayed in Figure 6.

A, D, and G of Figure 6 are cross-sectional views of an optical microscope, of in vitro and in vivo US findings of Allergan smooth. Its upper hyperechoic line shows a clear differentiation from a macrotextured breast implant. However, it is not easy to differentiate it from Motiva nanotexture, or Mentor smooth and Sebbin smooth.

B, E, and H of Figure 6 are cross-sectional views of an microscope, of in vitro US, and in vivo HRUS findings of Mentor smooth. Its upper hyperechoic line shows a clear differentiation from a macrotextured breast implant. However, it is not easy to differentiate it from Motiva nanotexture, Allergan smooth, and Sebbin smooth.

C, F, and I of Figure 6 are cross-sectional views of an optical microscope, of in vitro US findings and in vivo HRUS ones, respectively, of Sebbin smooth. Its upper hyperechoic line shows a clear differentiation from a macrotextured breast implant. Furthermore, it is not relatively easier to differentiate it from Motiva nanotexture, smooth of Mentor and Allergan. 

## 8. Discussion

Since the emergence of BIA-ALCL, the number of patients undergoing breast implant examinations has increased in Korea. However, most of these patients are unaware of the information about their implants [26]. 

Moreover, breast implant-associated squamous cell cancer after silicone gel breast implantation surgery was reported, too. [27,28,29,30]. 

Mammogram, US, and MRIs are available diagnostic imaging modalities for monitoring breast implant-related complications, such as ruptures, seromas, and capsular contractures.

These imaging modalities are not used for identifying the shell type of breast implants. 

However, an immediate, reliable method for identifying the shell type of a breast implant is essential. Evidence-based research and the applying real-world techniques that identify the surface topographic information without surgery have become of paramount importance for breast physicians. 

This study is unique in that it connects the physical properties of an implant to in vivo US images. There is a variability in the classification system proposed by fundamental metrology. It also remains problematic that there is no match between the classification system used by the manufacturer and fundamental metrology [7]. In a clinical setting, however, the surface topography can be classified into macro-, micro- and nanotextured and smooth surfaces [12,31].

Many confusing texture scale classifications measured by various devices such as interferometry, SEM and X-ray CT, micro-CT, and digital microscopy that are not available in private practices [3,5,25].

Atlan et al. reported that based on surface area characteristics and measurements, it is possible to group the textures into four classifications: smooth/nanotexture (80–100 mm^2^), microtexture (100–200 mm^2^), macrotexture (200–300 mm^2^) and macrotexture-plus (>300 mm^2^) [3].

Barr et al. proposed an implant classification system based upon statistical roughness, with a subcategorization based on re-entrance and surface roughness into four main types: macro-, micro-, meso- and nanotextured (smooth) [5].

According to the ISO14607 (nonactive surgical implants—mammary implants—particular requirements): in 2018, the surface was classified based on the formula, Ra = average surface roughness (height parameter variance between peaks and cavities along a defined profile length) [32].

The above studies have examined diverse types of evenness of surfaces, the presence of pores, the size and openness of the pores, and the depth of texturing [3].

The current study, we examined whether US would be efficient in differentiating between a macrotexture, microtexture, and smooth surface based on their cross-sectional view according to the conventional classification system.

As fundamental metrology has proposed a diverse types of classification systems using various equipment, including ultrasonography, it showed a good performance in identifying the four types of breast implant surface types (i.e., smooth, nanotexture, microtexture and macrotexture).

All smooth breast implants were easily differentiated from any macrotextured breast implants. However, the micro-textured and smooth-types implants had almost the same image pattern in US. So, it was hard to differentiate the smooth and microtexture breast implant types. 

Identifying the breast implant shell type by US is very useful for patients concerned about BIA-ALCL and lack of breast implant information. There were many cases of patients having no idea about their previous implant surgery information, such as shape, constituents, manufacturer, and shell type. These situations are often observed due to many private clinic closures in Korea.

## 9. Limitations

This study could not analyze all the breast implants that are commercially available in the global market. This study focused only on breast implants that were manufactured in or imported to Korea and approved by the KFDA. The study identified the accuracy of ultrasonography in limited patients. The result found in this study are hard to generalize because ultrasonography is operator dependent. 

## 10. Conclusions

There are diverse methods of classifying the surface of a breast implants. In a real-world setting, however, it is necessary to identify a macro-textured devices as there is a causal relationship with BIA-ALCL. In this regard, HRUS was found to be useful in differentiating between macrotextured and smooth breast implants, including microtextured or nanotextured devices based on the texture topography cross-sectional view. 

Although imperfect, HRUS is a useful diagnostic tool for distinguishing between macro-textured breast implants and smooth (micro/nanotexture)-type breast implants. 

## Figures and Tables

**Figure 1 medicina-59-01092-f001:**
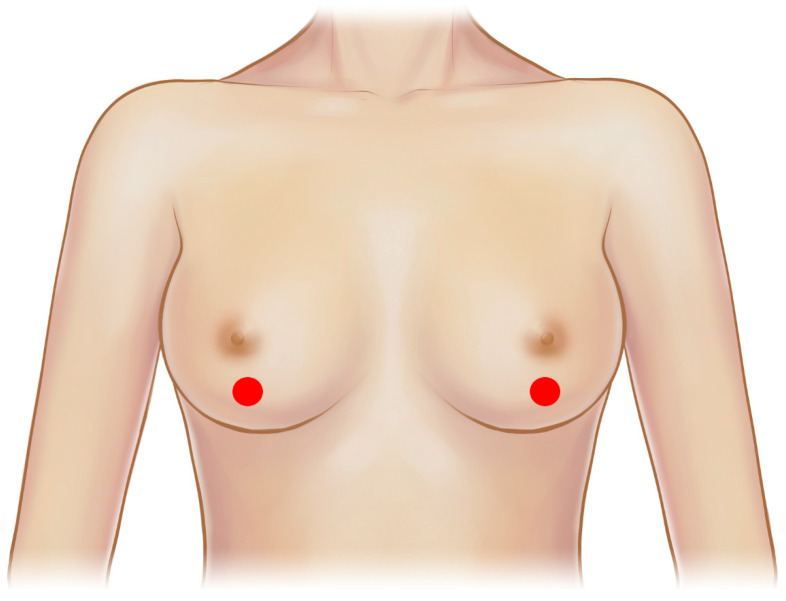
Ideal areas (red dots) for identifying the breast implant shell surface topography.

**Figure 2 medicina-59-01092-f002:**
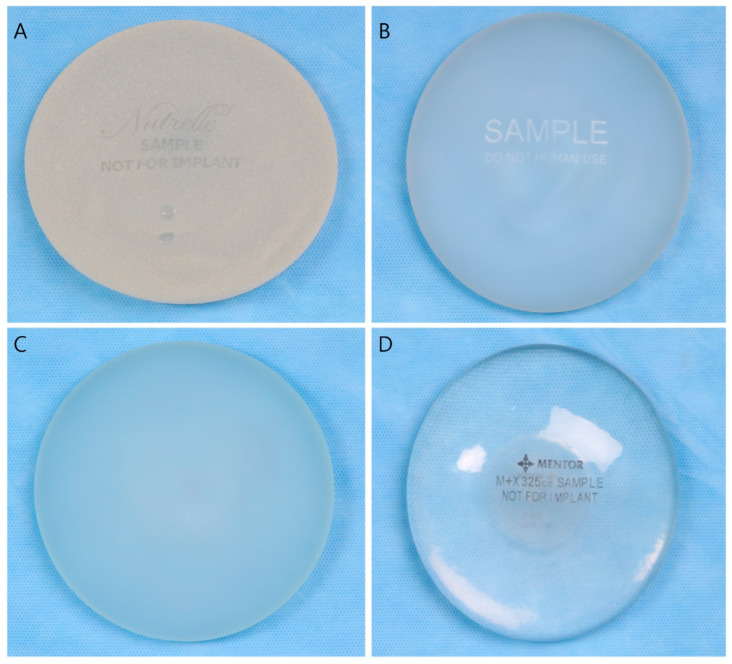
Various shell types of breast implants, gross view. (**A**) Macrotexture, (**B**) microtexture, (**C**) nanotexture, (**D**) smooth.

**Figure 3 medicina-59-01092-f003:**
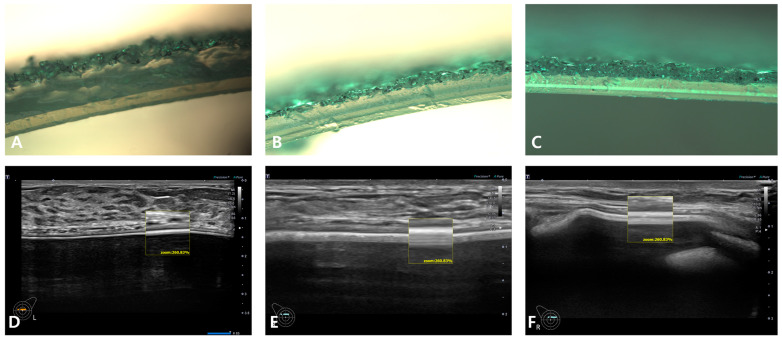
Microscopic and ultrasonographic cross-sectional views of three (3) embossed macrotexture-type breast implants. (**A**) Mentor, (**B**) Silimed, (**C**) Polytech. These are light microscopic cross-sectional images of embossed macrotexture-type breast implants ((**A**–**C**): ×400). (**D**) Mentor, (**E**) Silimed, (**F**) Polytech. These are ultrasonographic cross-sectional images of embossed macrotexture-type breast implants.

**Figure 4 medicina-59-01092-f004:**
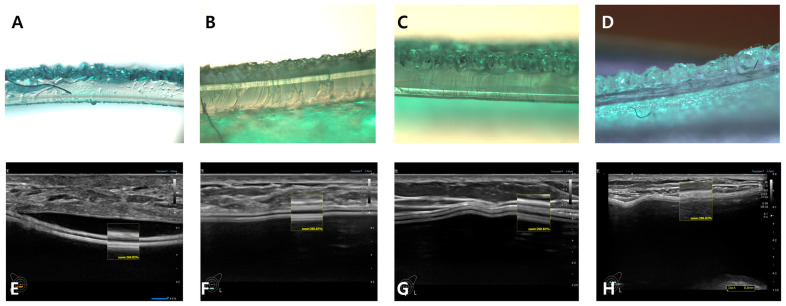
Microscopic and ultrasonographic cross-sectional view of four (4) engraving macrotexture-type breast implants (**A**–**D**: ×400). (**A**) Allergan, (**B**) Sebbin, (**C**) HansBiomed, (**D**) Eurosilicone. These are light microscopic cross-sectional images of engraved macrotexture-type breast implants (**A**–**D**: ×400). (**E**) Allergan, (**F**) Sebbin, (**G**) HansBiomed, (**H**) Eurosilicone. These are light microscopic cross-sectional images of engraving macrotexture-type breast implants.

**Figure 5 medicina-59-01092-f005:**
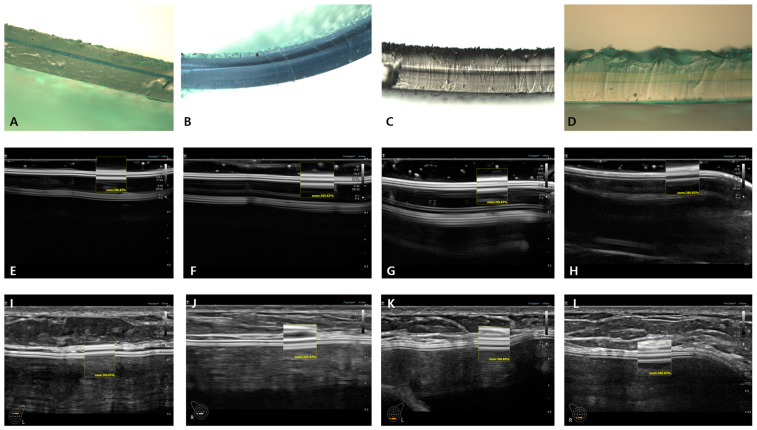
Microscopic and ultrasonographic cross-sectional views of three (3) microtexture- and one (1) nanotexture-type breast implants. (**A**) Motiva, (**B**) HansBiomed, (**C**) Sebbin, (**D**) Eurosilicone. These are light microscopic images of three (3) microtexture- and one (1) nanotexture-type breast implants. (**E**) Motiva, (**F**) HansBiomed, (**G**) Sebbin, (**H**) Eurosilicone. These are in vitro ultrasonographic cross-sectional images of three (3) microtexture- and one (1) nanotexture-type breast implants. (**I**) Motiva, (**J**) HansBiomed, (**K**) Sebbin, (**L**) Eurosilicone. These are in vitro ultrasonographic cross-sectional images of 3 microtexture- and 1 nanotexture-type breast implants.

**Figure 6 medicina-59-01092-f006:**
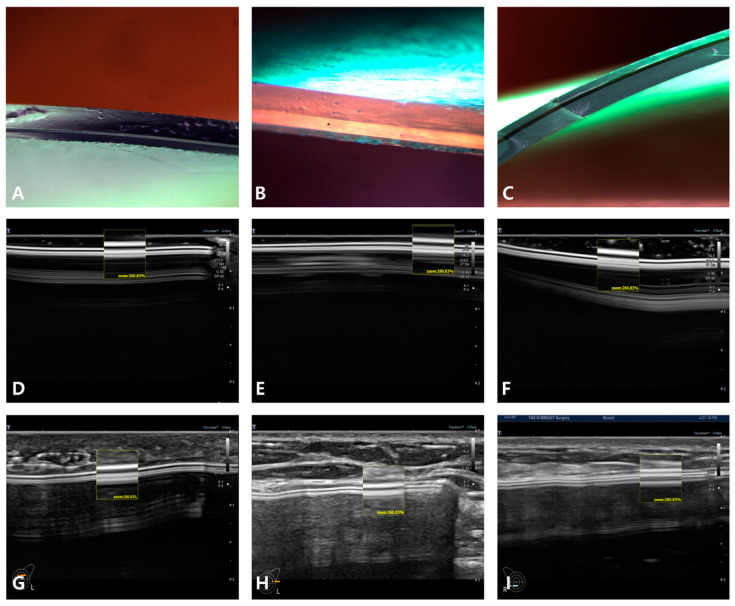
Microscopic and ultrasonographic cross-sectional views of three (3) smooth-type breast implants (**A**–**C**: ×400). (**A**) Allergan, (**B**) Mentor, (**C**) Sebbin. These are light microscopic images of three (3) smooth shell surface breast implants. (**D**) Allergan, (**E**) Mentor, (**F**) Sebbin. These are in vitro ultrasonographic images of three (3) smooth shell surface breast implants. (**G**) Allergan, (**H**) Mentor, (**I**) Sebbin. These are in vivo ultrasonographic images of three (3) smooth shell surface breast implants.

**Table 1 medicina-59-01092-t001:** Available shell surface types of breast implants from 8 manufacturers.

Manufacturer	Macrotextured	Microt Extured	Nanotextured/Smooth
Allergan, Irvine, CA, USA	O	N/A	O
Mentor Worldwide LLC, Santa Barbara, CA, USA	O	N/A	O
Silimed, Rio de Janeiro, Brazil	O	N/A	N/A
Sebbin SAS, Boissy-l’Aillerie, France	O	O	O
GC Aesthetic PLC, Apt Cedex, France	O	O	N/A
Polytech Health & Aesthetics, Dieburg, Germany	O	N/A	N/A
Motiva, Establishment Labs Holdings Inc., Alajuela, Costa Rica	N/A	N/A	O
HansBiomed, Seoul, Republic of Korea	O	O	N/A

**Table 2 medicina-59-01092-t002:** Characteristics of 1901 patients.

Variable	Value (Mean ± SD), (%)
Age (years old)	35.9 ± 8.3
Sex (male-to-female ratio)	0:1901
Height (cm)	162.7 ± 4.8
Weight (kg)	53.1 ± 6.4
BMI (kg/m^2^)	20.0 ± 2.1
Days from previous surgery	41,870.3 ± 2169.5
Less than 3 years	570 (30.0), 521.3 ± 317.3
3~10 years	908 (47.7), 2318.3 ± 720.3
10~20 years	361(19.0), 4760.7 ± 928.8
More than 20 years	47 (2.5), 10,534.5 ± 7480.0
Not applicable	15 (0.8)

**Table 3 medicina-59-01092-t003:** Characteristics of implant-based mammaplasties of 3802 breast implants.

Variable	Value (%)
Purpose of surgery	
Esthetic	3793 (99.8)
Reconstructive	9 (0.2)
Type of incision	
Trans-axillary	2503 (65.8)
IMF	959 (25.2)
Peri-areolar	277 (7.3)
Mastectomy scar	3 (0.1)
Umbilical	56 (1.5)
Other	4 (0.1)
Type of pocket	
Subpectoral	3524 (92.7)
Subglandular	278 (7.3)
Fill material	
Silicone	3615 (95.1)
Saline	181 (4.7)
Dual chamber	6 (0.2)
Shell type	
Texture	2034 (53.5)
Microtexture	73 (1.9)
Smooth	1622 (42.7)
Not applicable due to rupture	73 (1.9)
Shape type	
Anatomical	1011 (26.6)
Round	2642 (69.5)
Not applicable due to rupture	149 (3.9)
Manufacturer	
Sebbin	175 (4.6)
HansBiomed	24 (0.6)
Motiva	14 (0.4)
Eurosilicone	12 (0.3)
Allergan	1079 (28.4)
Mentor	236 (6.2)
Polytech	265 (7.0)
Silimed	96 (2.5)
Other	0 (0.0)
Unknown due to–roundness	1764 (46.4)
–Rupture	137 (3.6)

## Data Availability

No data were created in this study.

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
