# Peer review of "Use of High-Resolution Ultrasound in Characterizing the Surface Topography of a Breast Implant"

_medicina, 2023, doi:10.3390/medicina59061092_

Round 1

Reviewer 1 Report

As is mentioned in the Conclusion of this paper, there is a limitation to using or judging breast implant that is manufactured in Korea only, thus the results obtained are limited to those available in the local market of Korea, So we might have another point of view or might be concluded different reasoning in a case of using implant other than those used in this study.

Author Response

We greatly appreciate your comments on our manuscript.

We understand your skepticism regarding the availability of breast implant which is commercially available in Korea only. 

As you mentioned, the assertive conclusion has been changed as  your opinion. 

Conclusions

HRUS is a useful diagnostic modality for identifying the various surface shell type of breast implant. These shell type information would be helpful to patients who had no idea for previous implant surgery information with the concern of BIA-ALCL.

And we added the more references. 

Reviewer 2 Report

Kim et al. describe in their study the advantages of using high-resolution ultrasound in the characterization of breast implants. The tables and figures are good and  and representative for the subject. The study was conducted with good intentions, however there is key information missing in the manuscript, and also a need to perform moderate English changes.

  1. The abstract needs to be reformulated, especially the  Background and Objectives” section. Please explain all the abbreviations in the abstract.
  2. The Introduction can be extended to include more data about HRUS and BIA-ALCL (such as its main causes, its pathology, its ultrasound appearance, the impact it has on patients survival and quality of life etc.)
  3. I believe that the “Breast Implant shell topography” section from Materials and Methods can be moved in the Introduction, as part of breast implants’ description.
  4. Regarding the Results section, I suggest including both the tables and the figures inside the manuscript and not at the end of it, as it will make the paper more easy to read. Also, I suggest putting the paragraphs which detail the images as figure descriptions, with the same reasoning – it would make the manuscript more intelligible and the figures more easy to follow.
  5. Although the paper stars by stating the importance of analysing breast implants’ surface texture for early detection of BIA-ALCL, the results do not mention anything on this subject. Did the authors find any causality between the texture of the implants and the appearance of BIA-ALCL? Were there any patients in the analysed cohort that presented the disease? Was HRUS sensitive in early detection of the disease? Which of the implants associated a higher rate of BIA-ALCL appearance?
  6. The discussion section is weak and needs improvement.
  7. The limitations can be expanded as well.
  8. The number of references is too small – I suggest improving the manuscript with more data from the literature, having at least 30 cited references.

There are also a lot of grammar mistakes, such as “breast implant was” in the first sentence of the introduction (it should be “breast implants were”), “the number of patients that received the implant” in the fourth paragraph of the same section, just to mention a few. In addition to this, a lot of the sentences need rephrasing, for example the first section of the abstract (as mentioned before), the fifth paragraph from the results section (“Firstly, these are macro-texture topographic images…”) etc. 

Author Response

We greatly appreciate your comments on our manuscript.

We revised the manuscript and edited English. 

Reviewer 3 Report

The title of the article reflects the work. The purpose of the study is adequately explained in the article. The abstract adequately reflects the article. The article is in accordance with scientific ethics. The author should improve the rationale and clinical manifestation of macrotextured devices associated complications.

Nil

Author Response

We greatly appreciate your comments on our manuscript.

We did more editing of English language and added more references. 

Round 2

Reviewer 2 Report

The authors performed the changes according to my suggestions.

I do still believe that there is a dire need for English grammar changes. I suggest having a professional looking over your manuscript.